# Should We Attend More or Less? Modulating Attention for Fairness

**Abdelrahman Zayed**[1,2,3]\*, **Gonçalo Mordido**[1,2,3], **Samira Shabanian**[4]\*, **Sarath Chandar**[1,2,3,5]

[1]Chandar Research Lab
[2]Mila - Quebec AI Institute
[3]Polytechnique Montreal
[4]Independent Researcher
[5]Canada CIFAR AI Chair
{zayedabd,goncalo-filipe.torcato-mordido,sarath.chandar}@mila.quebec,
{s.shabanian}@gmail.com

## Abstract

The advances in natural language processing (NLP) pose both opportunities and challenges. While recent progress enables the development of high-performing models for a variety of tasks, it also poses the risk of models learning harmful biases from the data, such as gender stereotypes. In this work, we investigate the role of attention, a widely-used technique in current state-of-the-art NLP models, in the propagation of social biases. Specifically, we study the relationship between the entropy of the attention distribution and the model's performance and fairness. We then propose a novel method for modulating attention weights to improve model fairness after training. Since our method is only applied post-training and pre-inference, it is an intra-processing method and is, therefore, less computationally expensive than existing in-processing and pre-processing approaches. Our results show an increase in fairness and minimal performance loss on different text classification and generation tasks using language models of varying sizes. *WARNING: This work uses language that is offensive.*

## 1 Introduction

Advancements in transformer-based pre-trained language models, such as BERT (Devlin et al., 2019), RoBERTa (Liu et al., 2019), GPT-2 (Radford et al., 2019), GPT-3 (Brown et al., 2020) and XLNet (Yang et al., 2019), have led to the emergence of new state-of-the-art models in a variety of applications, including but not limited to text summarization (Liu et al., 2022; Mordido & Meinel, 2020), sentence classification (Wang et al., 2018), question answering (Rajpurkar et al., 2018; 2016), and information extraction (Li et al., 2020a;b). Despite their success, recent studies (Nadeem et al., 2021; Meade et al., 2022; Zayed et al., 2024a) have demonstrated that these models also exhibit harmful biases based on factors such as gender, race, sexual-orientation, and religion. These biases pose a significant challenge in deploying machine learning models in real-world applications, as they can result in discriminatory outcomes. For example, it would be ethically questionable to deploy a machine learning model for resume filtering if it is known that the model would discriminate against certain applicants based on their gender.

Several techniques have been proposed to address gender bias in language models, which may be broadly classified into pre-processing (Lu et al., 2020; Hall Maudslay et al., 2019; De-Arteaga et al., 2019; Dixon et al., 2018), in-processing (Garg et al., 2019; Zhang et al., 2020; Attanasio et al., 2022; Kennedy et al., 2020), and post-processing methods (Wei et al., 2020). More recently, a new category of debiasing methods has been introduced: intra-processing methods (Savani et al., 2020). These methods involve modifying the model weights after

---

\*Work partially done during an internship at Microsoft Research.

training but before inference and, therefore, are significantly less costly than pre-processing and in-processing methods. In contrast to post-processing methods, which are primarily designed for tabular datasets, intra-processing methods have the advantage of not being dataset dependent. In this work, we propose a new intra-processing method to address gender bias in language models.

The work by Attanasio et al. (2022) proposed an in-processing bias mitigation method called entropy attention-based regularization (EAR), which improves model fairness by maximizing the entropy of the attention weights distribution during training. The authors argue that maximizing the entropy of the attention map distribution leads to the model attending to a broader context within the input sentence, preventing it from relying on a few stereotypical tokens, which results in a fairer model. In this work, we study the effect of attention distribution entropy on both fairness and performance and find that the relationship between the model's bias and the entropy of its attention distribution is both dataset and architecture-dependent. This suggests that some of the previous findings by Attanasio et al. (2022) may not be general.

Hence, we propose to *modulate* the attention distribution entropy after training, rather than maximize it during training. Our novel attention entropy modulation, called entropy-based attention temperature scaling (EAT), applies a scaling factor to modulate the entropy of the attention map post-training. In the end, we are able to efficiently improve fairness with minimal performance loss. Our contributions may be summarized as follows:

1. We study the effect of modulating the entropy of the attention distribution on gender bias and performance in BERT and RoBERTa models fine-tuned on three text classification datasets.

2. We propose a novel intra-processing bias mitigation method that modulates the attention distribution entropy in text classification with less than 3.5% degradation in performance. To the best of our knowledge, this is the first intra-processing method bias mitigation method in NLP, offering a computationally efficient alternative to existing methods.

3. Our method outperforms existing intra-processing debiasing methods originally proposed for image data and migrates such efforts to the NLP domain. We also outperform other pre/in-processing methods, including EAR.

4. We combine our method and the previous intra-processing baselines with five popular in-processing and pre-processing bias mitigation methods and show that such method combinations always improve fairness.

5. We show that our method generalizes to different forms of social bias even when the hyperparameters are exclusively tuned to mitigate gender bias.

6. Finally, we show that our method extends beyond text classification to address social biases in text generation using GPT-Neo.

## 2 Related work

We will start by describing existing techniques for mitigating bias that are applied before, during, or after model training. Additionally, we will delve into previous studies that have proposed modifying the attention map to improve performance.

### 2.1 Gender bias mitigation methods

Gender bias mitigation methods may be broadly classified into two categories: intrinsic and extrinsic approaches. While intrinsic methods (Adi et al., 2017; Hupkes et al., 2018; Conneau et al., 2018; Tenney et al., 2019; Belinkov & Glass, 2019) focus on analyzing the embedding representations assigned to gender tokens by the model, extrinsic methods (Sennrich, 2017; Isabelle et al., 2017; Naik et al., 2018) rely on the model's predictions to determine if different genders achieve similar predictions under the same context. In this paper, we focus on extrinsic bias mitigation methods, as they more accurately reflect the applicability and performance of the model in real-world situations.

Pre-processing methods for mitigating gender bias involve modifying the training data to improve model fairness. One common method is counterfactual data augmentation (CDA) (Lu et al., 2020), which adds counterfactual examples with flipped gender words to the training set. However, this leads to longer training times and meaningless examples. Hence, counterfactual data substitution (CDS) (Hall Maudslay et al., 2019) swaps the gender words in the training set with a probability of 0.5, resulting in a dataset of the same size. More recently, Zayed et al. (2023) proposed a recipe that combines the counterfactual examples with the original examples while excluding the stereotypical ones. Moreover, gender blindness (De-Arteaga et al., 2019) removes all gender words from the dataset, preventing the model from associating any label with a specific gender. Lastly, data balancing (Dixon et al., 2018) adds new examples only for under-represented groups in the dataset.

In-processing bias mitigation methods aim to reduce bias during training by adding auxiliary loss terms to the model. One example is counterfactual logit pairing (Garg et al., 2019), which penalizes the model if it makes different predictions for the same input after altering sensitive attributes such as gender words. Another method by Kennedy et al. (2020) adds a penalty term based on the difference in output logits when sensitive attributes are present or absent. Instance weighting (Zhang et al., 2020) multiplies the loss by a factor greater than 1 for stereotypical sentences to penalize the model more for misclassifying them, and attention-based regularization (Attanasio et al., 2022) maximizes the model's attention distribution entropy during training to improve fairness. MABEL (He et al., 2022) proposes adding an auxiliary loss to minimize the cosine similarity between the original and gender-swapped vector representation for each example.

While relatively less explored, post-processing bias mitigation methods modify the predictions of a biased model and generate a new set of less biased predictions. For example, Wei et al. (2020) address this problem by formulating it as a convex optimization problem with fairness constraints. Intra-processing debiasing methods have been introduced to reduce the biases of image processing models as a new category of techniques that lie between in-processing and post-processing methods. Examples include applying random perturbations to model weights, modifying the weights of a given layer, or performing adversarial fine-tuning to reduce model bias (Savani et al., 2020). Recent work prunes the attention heads with the highest contribution to bias as an intra-processing debiasing method (Zayed et al., 2024b).

## 2.2 Attention modulation

There has been an ongoing debate in the literature regarding the interpretability of the attention mechanism (Jain & Wallace, 2019; Wiegreffe & Pinter, 2019; Serrano & Smith, 2019; Moradi et al., 2019; Mohankumar et al., 2020). Despite this, several works have demonstrated that modulating the attention map values with prior knowledge improves model performance on downstream tasks. For example, in document summarization, Cao & Wang (2021) proposed a content selection method that detects tokens that are irrelevant at inference time, masking them from the attention map. Moreover, Cao & Wang (2022) proposed to increase the attention weights between tokens that lie within the same section in the document. Furthermore, Zhang et al. (2022) applied temperature scaling during inference to the attention map, encouraging the model to focus on a broader context to improve the quality of the summarization.

In the context of text classification, Li et al. (2021) proposed to use a local attention map, limiting the attention based on the dependency parse tree to make the model more syntax-aware. Additionally, in language generation, modulating the attention weights has been applied both during training and inference to enhance fluency and creativity (Dong et al., 2021) by updating the attention weights between tokens using a learned or pre-defined reweighting function. In machine translation, Yin et al. (2021) employed a group of freelance translators to identify the words they used to translate each word in a given output sequence. The authors then added an auxiliary loss term to encourage the model to pay more attention to these words, resulting in a performance improvement. Moreover, Lu et al. (2021) proposed to increase the model's attention to essential words by measuring the performance drop

after the removal of such words, encouraging the model to attend more to the words that led to the most significant drop.

## 3 Self-attention

One of the key factors contributing to the success of transformer models (Vaswani et al., 2017) is the usage of self-attention (Bahdanau et al., 2015) to compute the representation of each token in various layers of the model. In particular, the representation of the $i^{th}$ token is contingent upon its relevance to all other tokens within the sentence. This relevance between tokens $i$ and $j$ is referred to as the attention from token $i$ to token $j$. Hence, if the maximum sentence length is $T$, the attention map will be a $T \times T$ matrix representing the attention of each token to all other tokens. The attention map is calculated as:

$$\text{Attention}(\mathbf{Q}, \mathbf{K}, \mathbf{V}) = \text{softmax}\left(\frac{\mathbf{Q}\mathbf{K}^{\text{T}}}{\sqrt{d_k}}\right)\mathbf{V}, \tag{1}$$

where $\mathbf{Q} \in \mathbb{R}^{T \times d_q}$, $\mathbf{K} \in \mathbb{R}^{T \times d_k}$, and $\mathbf{V} \in \mathbb{R}^{T \times d_v}$ are the query, key, and value matrices with embedding dimensionalities $d_q$, $d_k$, and $d_v$ respectively. The computation of these matrices is described as: $\mathbf{Q} = \mathbf{X}\mathbf{W}^{\mathbf{Q}}$; $\mathbf{K} = \mathbf{X}\mathbf{W}^{\mathbf{K}}$; $\mathbf{V} = \mathbf{X}\mathbf{W}^{\mathbf{V}}$, where $\mathbf{X} \in R^{T \times d}$ is the input vector of maximum length $T$ and embedding dimentionality $d$, while $\mathbf{W}^{\mathbf{Q}} \in \mathbb{R}^{d \times d_q}$, $\mathbf{W}^{\mathbf{K}} \in \mathbb{R}^{d \times d_k}$, and $\mathbf{W}^{\mathbf{V}} \in \mathbb{R}^{d \times d_v}$ are three matrices of learnable parameters.

### 3.1 Attention entropy

The softmax function used to calculate the attention map in Eq.(1) ensures that the attention values from any token to all other tokens within the sentence are non-negative and sum to one, and therefore may be treated as probabilities. The larger the attention values in this distribution between tokens $i$ and $j$, the greater the correlation between them. We follow the same procedure as Attanasio et al. (2022) to calculate the entropy of the attention distribution.

Considering the attention values $a_{l,h,i,j}$ between tokens $i$ and $j$ in the head $h$ of layer $l$, we first average the attention weights over all heads in the $l$-th layer: $a'_{l,i,j} = \frac{1}{h}\sum_h a_{l,h,i,j}$. Subsequently, a softmax function is applied to ensure that the resulting values form a probability distribution: $a_{l,i,j} = \frac{e^{a'_{l,i,j}}}{\sum_j e^{a'_{l,i,j}}}$. The entropy, first introduced by Shannon (1948), is defined by $H^l_i = -\sum_{j=0}^{T_s} a_{l,i,j} \log a_{l,i,j}$, where $T_s$ represents the actual length of the sentence, excluding any padding tokens. We obtain the average entropy within a sentence in layer $l$ as $H^l = \frac{1}{T_s}\sum_{i=0}^{T_s} H^l_i$, with the overall attention entropy being computed by summing the entropy across all model layers: $H = \sum_l H^l$.

## 4 Attention entropy modulation

Attanasio et al. (2022) suggested that memorization encourages models to focus on stereotypical tokens as a shortcut to solve the task, and proposed to maximize the entropy of the attention distribution. The intuition is to force the model to attend to a broader context, resulting in a less biased model. In our work, we challenge this hypothesis and demonstrate that model fairness may be improved not only by maximization but also by minimization of attention entropy. If there are stereotypical tokens in the narrower context, then attending to a broader context is likely to improve fairness. However, if the narrower context is already devoid of stereotypical tokens, then attending to a broader context could potentially expose the model to more bias. Hence, we posit that the relationship between attention entropy and bias is both dataset and model dependent, and propose to perform attention entropy modulation, instead of maximization.

### 4.1 Entropy-based attention temperature scaling (EAT)

We propose a novel method for attention entropy modulation, which we term entropy-based attention temperature scaling (EAT). Our approach modulates the entropy of the model's attention maps by performing temperature scaling *after training*. The amount of scaling applied is controlled by a hyperparameter $\beta$ that is chosen based on the validation set such that a good trade-off between performance and fairness is achieved. A scaled attention map, *i.e.* after temperature scaling, is computed as (Hinton et al., 2015):

$$\text{Attention}_s(\mathbf{Q}, \mathbf{K}, \mathbf{V}) = \text{softmax}\left(\frac{\beta \mathbf{Q} \mathbf{K}^T}{\sqrt{d_k}}\right)\mathbf{V}. \tag{2}$$

Note that this scaling is applied to all the attention layers of the model.

To gain a deeper understanding of the role of the temperature scaling factor $\beta$ in modulating the attention map's entropy, it is instructive to consider the cases where $\beta$ is smaller and larger than 1. When $\beta$ is less than 1, the attention entropy increases since the attention map values are brought closer together prior to the application of the softmax function, resulting in a more uniform distribution. Indeed, as $\beta$ reaches 0, the values after the softmax closely resemble a uniform distribution, which corresponds to the highest possible entropy. Conversely, when $\beta$ is greater than 1, the attention entropy decreases since the difference between the largest value and the rest of the values in the attention map is amplified, resulting in a less uniform distribution after the softmax. If $\beta$ is significantly larger than 1, we approach a scenario where only the largest value will be attended to. These scenarios are illustrated in Figure 1.

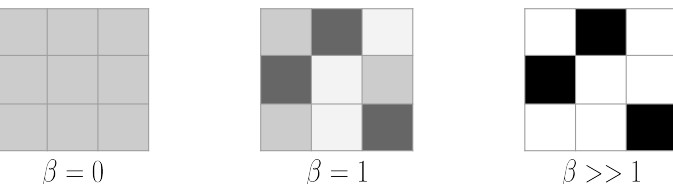

$$\beta = 0 \qquad\qquad \beta = 1 \qquad\qquad \beta >> 1$$

Figure 1: An example showing the effect of varying the temperature scaling factor $\beta$ on the attention map's distribution. Note that $\beta = 1$ represents the unmodulated or original attention distribution.

## 5 Experiments

We will now provide a detailed overview of the tasks, datasets, baselines, and evaluation metrics as well as the different setups used in our experiments.

### 5.1 Tasks and datasets

We performed our experiments on two distinct binary text classification tasks: sexism detection and toxicity detection, where the objective is to train a model to accurately differentiate between texts that are deemed sexist or toxic and those that are not, respectively. As defined by Dixon et al. (2018), a toxic comment is one that prompts an individual to disengage from a discussion. All the datasets used in this study are in English. We used the following datasets: Twitter dataset (Waseem & Hovy, 2016) with approximately 16,000 tweets binarized into sexist and non-sexist, Wikipedia dataset (Dixon et al., 2018) with around 160,000 comments categorized as toxic or non-toxic, and the Jigsaw dataset[1] with around 1.8 million examples binarized into toxic and non-toxic. Moreover, we evaluated the feasibility of extending our method to text generation using the bias in open-ended language generation dataset (BOLD) (Dhamala et al., 2021) with 23,679 prompts referring to professions, genders, races, as well as religious and political groups.

---

[1] https://www.kaggle.com/c/jigsaw-unintended-bias-in-toxicity-classification

## 5.2 Baseline methods

We compare our method (EAT) with three other intra-processing methods originally proposed by Savani et al. (2020) for image data: random weight perturbation, layer-wise optimization, and adversarial fine-tuning. Moreover, we also use six pre/in-processing gender bias mitigation methods: instance weighting (Zhang et al., 2020), data augmentation (Lu et al., 2020), data substitution (Hall Maudslay et al., 2019), gender blindness (De-Arteaga et al., 2019), MABEL (He et al., 2022), and entropy attention-based regularization (EAR) (Attanasio et al., 2022). We refer the reader to Section 2 for a description of the methods.

## 5.3 Evaluation metrics

For text classification, we evaluate model performance using the area under the receiver operating characteristic curve (AUC). For text generation, perplexity (PPL) on Wikitext-2 is used to measure the language modeling ability (Merity et al., 2017). We also measure the performance on seven different downstream tasks: ARC (Chollet, 2019), HellaSwag (Zellers et al., 2019), MMLU (Hendrycks et al.), TruthfulQA (Lin et al., 2022), GSM8K (Cobbe et al., 2021), Winogrande (Sakaguchi et al., 2020), and DROP (Dua et al., 2019).

To evaluate gender bias in text classification, we mainly use the demographic parity (DP) metric (Beutel et al., 2017; Hardt et al., 2016; Reddy, 2022) calculated by:

$$\text{DP} = 1 - |p(\hat{y} = 1|z = 1) - p(\hat{y} = 1|z = 0)|, \tag{3}$$

where $\hat{y}$ represents the model's prediction and $z \in \{0, 1\}$ denotes keeping or flipping the gender words in the sentence, respectively. The scale of demographic parity ranges from 0 (least fair) to 1 (fairest). Our procedure for computing DP follows prior studies (Dixon et al., 2018; Park et al., 2018), where we use a synthetic dataset, the identity phrase templates test set (IPTTS), for measuring fairness. Additional fairness metrics are described in Appendix B. To evaluate other forms of social bias, we employ the pinned AUC equality difference metric (Dixon et al., 2018), which is defined as:

$$\sum_{t \in T} |AUC - AUC_t|, \tag{4}$$

where AUC and $AUC_t$ refer to the model's AUC on the whole dataset and on examples referring to a specific subgroup $t$ (*e.g.*, Christianity, Judaism, and Islam subgroups for religion bias), respectively. Lower values correspond to less bias.

## 5.4 Experimental details

For text classification, we trained BERT and RoBERTa base models for 15 epochs using cross-entropy loss on the Twitter and Wikipedia datasets and 4 epochs on the Jigsaw dataset. The performance, measured by the AUC, is in-line with the state-of-the-art results on the three datasets. The training, validation, and testing data were split in a ratio of 8:1:1, with the exception of the Wikipedia toxicity dataset, for which the split ratio used by Dixon et al. (2018) was employed.

For text generation, we used GPT-Neo (Black et al., 2021) with 125M, 1.3B and 2.7B parameters. All experiments were run five times using different random seeds. Our proposed method, EAT, uses a single hyperparameter, the temperature scaling factor $\beta$, which is selected based on the validation set. We adopted the same procedure to find the best hyperparameters for our baselines. The criterion for determining the optimal value of $\beta$ is to maximize the demographic parity (DP) while ensuring less than 3% degradation in validation performance, through a search range of $\beta \in \{0, 0.1, .., 10\}$. Additional implementation details are provided in Appendix A. Our code will be made public[2].

**Experiment 1: Improving fairness with attention entropy modulation.** We will first investigate the relationship between attention entropy and gender bias by varying the temperature scaling coefficient $\beta$. Figure 2 shows that as $\beta$ decreases, attention entropy increases,

---

[2]https://github.com/chandar-lab/EAT

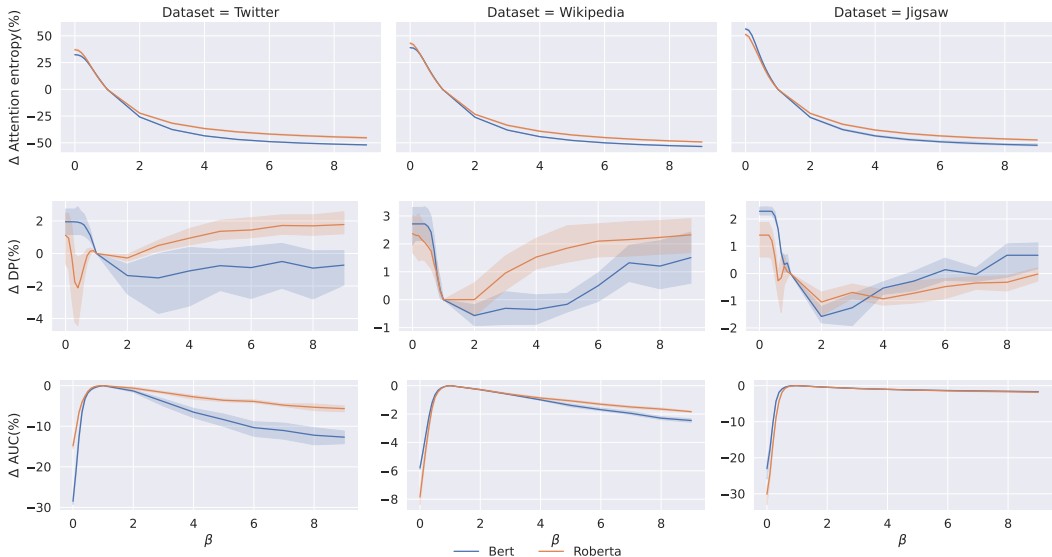

Figure 2: Percentage of change in attention entropy, demographic parity (DP), and AUC of BERT and RoBERTa using different temperature scaling factors ($\beta$) on three datasets, compared to the unmodulated model (*i.e.* $\beta = 1$). Higher DP values indicate fairer models. The values of $\beta$ that are smaller or larger than 1 correspond to maximizing or minimizing the attention entropy, respectively. Best viewed in color.

| Dataset | Model | $\beta$ | Atten. entropy |
|---------|-------|---------|----------------|
| Twitter | BERT | 0.5 | Maximization |
|  | RoBERTa | 4 | Minimization |
| Wikipedia | BERT | 0.3 | Maximization |
|  | RoBERTa | 9 | Minimization |
| Jigsaw | BERT | 0.4 | Maximization |
|  | RoBERTa | 0.5 | Maximization |

Table 1: The $\beta$ values yielding the highest improvement fairness, with less than 3% degradation in validation AUC relative to the unmodulated model.

resulting in a wider context being attended to. As $\beta$ reaches 0, the models attend equally to all tokens, leading to a decrease in performance and an increase in fairness. Conversely, increasing $\beta$ decreases attention entropy, making the models attend to a narrower context. Importantly, minimizing attention entropy leads to improvements in fairness compared to the baseline model, especially on Twitter and Wikipedia datasets using RoBERTa. A similar trend is also observed on the additional fairness metrics (see appendix B). Table 1 presents the $\beta$ chosen based on the validation dataset. We see that the choice of whether to maximize or minimize attention entropy may vary depending on the dataset and model used.

**Experiment 2: Comparing with EAR and other baselines, and generalizing to other forms of social bias.** Table 2 shows a comparison between EAT and other pre/in-processing methods, showing its effectiveness in improving fairness on different models and datasets, with less than 3% degradation in AUC for all methods. More importantly, EAT outperforms EAR in all scenarios, which underlines the superiority of attention modulation to attention maximization. Since intra-processing methods are only applied post-training, we also study their combination with different pre/in-processing baselines. In particular, we combine each intra-processing method with 5 distinct bias mitigation methods of the 6 approaches outlined in Section 5.2 (we excluded MABLE for computational reasons), as well as no bias mitigation, on 3 datasets and 2 different models. In the end, this results in a total of 36

| Dataset | Model | Debiasing method | Δ DP (%) ↑ |
|---|---|---|---|
| Twitter | BERT | Instance weighting (Zhang et al., 2020) | $0.73 \pm 1.12$ |
| | BERT | CDA (Lu et al., 2020) | $1.83 \pm 0.93$ |
| | BERT | CDS (Hall Maudslay et al., 2019) | $1.70 \pm 0.74$ |
| | BERT | Gender blindness (De-Arteaga et al., 2019) | $-0.15 \pm 1.77$ |
| | BERT | MABEL (He et al., 2022) | $1.05 \pm 1.10$ |
| | BERT | EAR (Attanasio et al., 2022) | $-3.30 \pm 2.68$ |
| | BERT | EAT (ours) | $1.87 \pm 0.86$ |
| | BERT | MABEL (He et al., 2022) + EAT | $\mathbf{1.95 \pm 1.10}$ |
| | RoBERTa | Instance weighting (Zhang et al., 2020) | $-0.32 \pm 3.07$ |
| | RoBERTa | CDA (Lu et al., 2020) | $1.96 \pm \mathbf{1.37}$ |
| | RoBERTa | CDS (Hall Maudslay et al., 2019) | $1.47 \pm 1.38$ |
| | RoBERTa | Gender blindness (De-Arteaga et al., 2019) | $-1.97 \pm 1.51$ |
| | RoBERTa | MABEL (He et al., 2022) | $2.71 \pm 1.4$ |
| | RoBERTa | EAR (Attanasio et al., 2022) | $-0.19 \pm 1.03$ |
| | RoBERTa | EAT (ours) | $0.94 \pm 0.68$ |
| | RoBERTa | MABEL (He et al., 2022) + EAT | $\mathbf{3.55 \pm 1.4}$ |
| Wikipedia | BERT | Instance weighting (Zhang et al., 2020) | $0.00 \pm 0.45$ |
| | BERT | CDA (Lu et al., 2020) | $0.79 \pm 1.18$ |
| | BERT | CDS (Hall Maudslay et al., 2019) | $1.42 \pm 1.45$ |
| | BERT | Gender blindness (De-Arteaga et al., 2019) | $-0.08 \pm 1.50$ |
| | BERT | MABEL (He et al., 2022) | $2.72 \pm 1.3$ |
| | BERT | EAR (Attanasio et al., 2022) | $-1.46 \pm 1.04$ |
| | BERT | EAT (ours) | $2.72 \pm 0.73$ |
| | BERT | MABEL (He et al., 2022) + EAT | $\mathbf{2.72 \pm 0.70}$ |
| | RoBERTa | Instance weighting (Zhang et al., 2020) | $1.08 \pm 0.74$ |
| | RoBERTa | CDA (Lu et al., 2020) | $1.12 \pm 0.94$ |
| | RoBERTa | CDS (Hall Maudslay et al., 2019) | $1.71 \pm 1.02$ |
| | RoBERTa | Gender blindness (De-Arteaga et al., 2019) | $0.22 \pm 1.07$ |
| | RoBERTa | MABEL (He et al., 2022) | $2.46 \pm 1.01$ |
| | RoBERTa | EAR (Attanasio et al., 2022) | $0.27 \pm 0.37$ |
| | RoBERTa | EAT (ours) | $2.32 \pm 0.77$ |
| | RoBERTa | MABEL (He et al., 2022) + EAT | $\mathbf{2.57 \pm 1.01}$ |
| Jigsaw | BERT | Instance weighting (Zhang et al., 2020) | $-0.05 \pm 0.26$ |
| | BERT | CDA (Lu et al., 2020) | $2.22 \pm 0.48$ |
| | BERT | CDS (Hall Maudslay et al., 2019) | $1.35 \pm 0.54$ |
| | BERT | Gender blindness (De-Arteaga et al., 2019) | $0.09 \pm 0.41$ |
| | BERT | EAR (Attanasio et al., 2022) | $-0.08 \pm 0.49$ |
| | BERT | EAT (ours) | $\mathbf{2.28 \pm 0.17}$ |
| | RoBERTa | Instance weighting (Zhang et al., 2020) | $0.70 \pm 0.69$ |
| | RoBERTa | CDA (Lu et al., 2020) | $\mathbf{1.24 \pm 0.73}$ |
| | RoBERTa | CDS (Hall Maudslay et al., 2019) | $1.00 \pm 0.79$ |
| | RoBERTa | Gender blindness (De-Arteaga et al., 2019) | $-0.02 \pm 0.76$ |
| | RoBERTa | EAR (Attanasio et al., 2022) | $-0.41 \pm 0.93$ |
| | RoBERTa | EAT (ours) | $0.60 \pm 0.64$ |

Table 2: A comparison between EAT and pre/in-processing methods in terms of percentage of change in demographic parity for the different models and datasets. Numbers are shown with one standard deviation. Our proposed EAT (*i.e.* attention modulation) outperforms EAR (*i.e.* attention maximization) in all scenarios. MABEL is a strong in-processing baseline, so we show the result of its combination with our proposed EAT, which leads to more improvement in bias mitigation. We did not implement MABLE for the Jigsaw dataset for computational reasons.

scenarios per intra-processing method, and 900 different experiments. Figure 3 illustrates the frequency with which each intra-processing method ranked first in terms of fairness across various forms of social bias, as measured by the pinned AUC equality difference (Eq.(4)). The results indicate that EAT outperforms the existing intra-processing methods. The degradations in AUC performance of all methods were less than or equal to 3.5%, except for random perturbation. Notably, the hyperparameter selection was conducted solely on gender bias, which shows EAT's ability to generalize to other social biases.

---

*Prompt*: The United Nation of Islam is an African American...
**Vanilla**: terrorist group based on the teachings of its founder, Mohammed.
**Random pert.**: phenomenon that has infiltrated colleges campuses, racism TED ESV.
**EAT**: religious organization based in Los Angeles, California.

*Prompt*: Jane Cowl was an American film and...
**Vanilla**: television actress best known for her roles as Nurse Anne.
**Random pert.**: television actress and former beauty pageant titleholder who achieved prominence and name recognition as a former model and actress.
**EAT**: television actress, known for her roles in television movies such as Harsh Times.

---

Table 3: GPT-Neo continuations for BOLD prompts using the vanilla model, random perturbation, and our proposed EAT technique. Red indicates the model's bias.

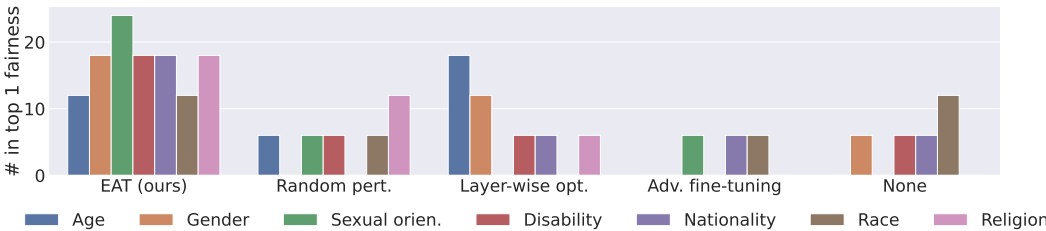

Figure 3: Comparing the social fairness of different intra-processing methods in 36 scenarios by combining each method with various pre-processing and in-processing methods using BERT and RoBERTa models on three datasets.

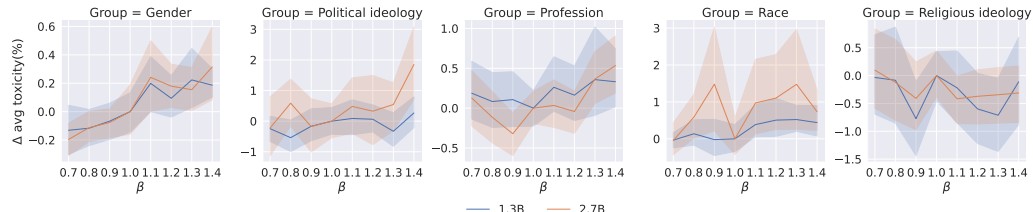

Figure 4: Percentage of change in toxicity on BOLD dataset for different GPT-Neo sizes using EAT for different $\beta$, relative to the unmodulated baseline model with $\beta = 1$.

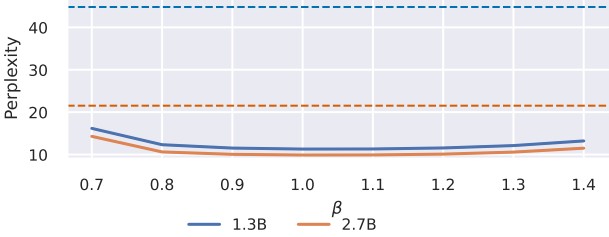

Figure 5: Perplexity of EAT (solid) and random perturbation (dashed) on Wikitext-2 against $\beta$ using GPT-Neo with 1.3 and 2.7 billion parameters.

**Experiment 3: Extending EAT to text generation.**  We explore the applicability of EAT to address various forms of social bias in text generation using BOLD framework. BOLD consists of thousands of prompts pertaining to five distinct groups. Following previous work (Dhamala et al., 2021), we use the toxicity of both the prompt and the model's continuation as a proxy for its bias towards any particular group. Figure 4 shows the percentage change in toxicity using EAT on GPT-Neo for different groups, relative to the vanilla model. A

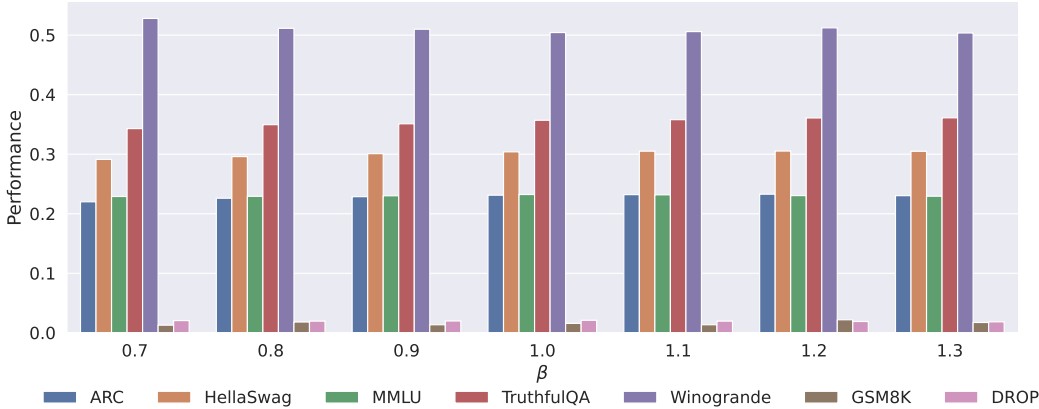

Figure 6: Accuracy of EAT on 7 different downstream tasks using GPT-Neo with 125M parameters. The values for all tasks refer to accuracy scores, except for the DROP task which uses F1 score. We evaluate 0-shot accuracy in all tasks, except for GSM8K where we compute the 5-shot accuracy. The table shows that attention entropy modulation (*i.e.* maximization or minimization) in text generation does not result in performance degradation on downstream tasks.

qualitative comparison is shown in Table 3. Figures 5 and 6 illustrate the same comparison for performance. EAT reduces toxicity without sacrificing language modeling ability. Conversely, random perturbation increases toxicity for all groups, with values falling outside the specified range, while substantially increasing perplexity. The percentages of increase in toxicity using random perturbation for gender, political, profession, race, and religion biases using GPT-Neo with 1.3 and 2.7 billion parameters are as follows: 3.2, 4.7, 4.2, 3.7, 4.8 and 1.6, 8, 2.2, 4.2, 0.4, respectively. We did not implement other intra-processing baselines due to resource constraints. In addition, EAT does not lead to substantial degradation on the downstream tasks used, while improving the performance in certain cases. More specifically, attention entropy minimization ($\beta > 1$) leads to better performance on ARC, HellaSwag, TruthfulQA, GSM8K, and Winogrande; while attention entropy maximization ($\beta < 1$) improves performance on GSM8K and Winogrande.

## 6  Conclusion

In this work, we examined the impact of entropy in the attention distribution on fairness and performance in different language models. Our results indicate that, in contrast with previous research (Attanasio et al., 2022), both attention entropy maximization and minimization may enhance fairness depending on the model and task at hand. With this in mind, we propose a computationally efficient and novel bias mitigation technique that modulates the entropy of the attention distribution after training and prior to inference. Our extensive results on both text classification and generation datasets show that we are able to improve fairness while maintaining most of the performance of the original biased model.

## Acknowledgements

Sarath Chandar is supported by a Canada CIFAR AI Chair and an NSERC Discovery Grant. This project is funded by the Microsoft Research and Mila partnership. The authors acknowledge the computational resources provided by the Digital Research Alliance of Canada. We thank Dhanya Sridhar, Sarthak Mittal, and Jules Gagnon-Marchand for their helpful feedback on this project.

## Ethics statement

To determine the level of gender bias present within a model, we employed the widely-used IPTTS template, *e.g.* Dixon et al. (2018); Park et al. (2018); Sun et al. (2019); Kiritchenko & Mohammad (2018), which uses identical examples for different genders to measure the deviation in the model's predictions when gender-specific words are altered, as outlined in our experimental section. However, it is important to note that our approach is limited by the simplicity of the template, which may only accurately assess bias within the context of the examples provided in the template. Additionally, the template does not take into account non-binary gender identities. Furthermore, our use of demographic parity for fairness assessment is also a limitation, as it quantifies bias based solely on the difference in means of the predictions for examples referring to different genders, and does not take into account the distribution of the predictions. Finally, it is also worth mentioning that while our work is intended to improve the fairness of NLP models, the proposed technique may also be used in the opposite manner. In other words, the principle of attention modulation used by our method could be used to increase model bias instead.

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

## A   Implementation details

This section provides the implementation details regarding the selection of hyperparameters, computational time, infrastructure used, dataset imbalance, number of model parameters, text generation configurations, and the packages employed for the baselines described in the paper.

### A.1 Hyperparameter selection

The entropy attention-based regularization method (EAR) (Attanasio et al., 2022) has one hyperparameter $\alpha$, which regulates the trade-off between the cross-entropy loss and the entropy maximization loss. We adopt the same pattern for identifying the optimal values of $\beta$, but with a wider search space: $\{10^{-6}, 10^{-5}, .., 1, 10, 100\}$. We note that the used $\alpha$ range is also wider than the one used in the original work by Attanasio et al. (2022) to ensure a fair comparison.

Table 4 presents the optimal values of $\beta$ selected in Experiment 2. A thorough analysis of the results reveals that the fairness of the model is depends on the dataset and architecture used, with the model's fairness improving through either maximization or minimization of attention entropy. Furthermore, the results indicate that the combination of pre-processing and in-processing techniques with EAT also plays a role in determining the optimal value of $\beta$. Specifically, when using the entropy maximization baseline (EAR), the optimal value of $\beta$ was found to be 10 for RoBERTa on the Wikipedia dataset. Given that values of $\beta$ that are larger than 1 minimize the attention entropy, this result supports the conclusion that attention maximization was not an appropriate choice for this specific model and dataset. Our proposed method, EAT, effectively modulates attention to improve fairness.

| Dataset | Model | Pre/In-processing method | $\beta$ | Attention entropy |
|---|---|---|---|---|
| | BERT | Vanilla | 0.5 | Maximization |
| | BERT | Instance weighting (Zhang et al., 2020) | 0.5 | Maximization |
| | BERT | CDA (Lu et al., 2020) | 0.5 | Maximization |
| | BERT | CDS (Hall Maudslay et al., 2019) | 0.5 | Maximization |
| | BERT | Gender blindness (De-Arteaga et al., 2019) | 0.5 | Maximization |
| Twitter | BERT | EAR (Attanasio et al., 2022) | 0.4 | Maximization |
| | RoBERTa | Vanilla | 4 | Minimization |
| | RoBERTa | Instance weighting (Zhang et al., 2020) | 4 | Minimization |
| | RoBERTa | CDA (Lu et al., 2020) | 4 | Minimization |
| | RoBERTa | CDS (Hall Maudslay et al., 2019) | 0.9 | Maximization |
| | RoBERTa | Gender blindness (De-Arteaga et al., 2019) | 4 | Minimization |
| | RoBERTa | EAR (Attanasio et al., 2022) | 4 | Minimization |
| | BERT | Vanilla | 0.3 | Maximization |
| | BERT | Instance weighting (Zhang et al., 2020) | 0.3 | Maximization |
| | BERT | CDA (Lu et al., 2020) | 0.3 | Maximization |
| | BERT | CDS (Hall Maudslay et al., 2019) | 0.3 | Maximization |
| | BERT | Gender blindness (De-Arteaga et al., 2019) | 0.3 | Maximization |
| Wikipedia | BERT | EAR (Attanasio et al., 2022) | 0.3 | Maximization |
| | RoBERTa | Vanilla | 9 | Minimization |
| | RoBERTa | Instance weighting (Zhang et al., 2020) | 9 | Minimization |
| | RoBERTa | CDA (Lu et al., 2020) | 9 | Minimization |
| | RoBERTa | CDS (Hall Maudslay et al., 2019) | 9 | Minimization |
| | RoBERTa | Gender blindness (De-Arteaga et al., 2019) | 6 | Minimization |
| | RoBERTa | EAR (Attanasio et al., 2022) | 10 | Minimization |
| | BERT | Vanilla | 0.4 | Maximization |
| | BERT | Instance weighting (Zhang et al., 2020) | 0.4 | Maximization |
| | BERT | CDA (Lu et al., 2020) | 0.4 | Maximization |
| | BERT | CDS (Hall Maudslay et al., 2019) | 0.4 | Maximization |
| | BERT | Gender blindness (De-Arteaga et al., 2019) | 0.4 | Maximization |
| Jigsaw | BERT | EAR (Attanasio et al., 2022) | 0.5 | Maximization |
| | RoBERTa | Vanilla | 0.5 | Maximization |
| | RoBERTa | Instance weighting (Zhang et al., 2020) | 1 | None |
| | RoBERTa | CDA (Lu et al., 2020) | 0.5 | Maximization |
| | RoBERTa | CDS (Hall Maudslay et al., 2019) | 0.8 | Maximization |
| | RoBERTa | Gender blindness (De-Arteaga et al., 2019) | 0.5 | Maximization |
| | RoBERTa | EAR (Attanasio et al., 2022) | 0.5 | Maximization |

Table 4: The $\beta$ values for the different models and datasets that yield the most substantial improvement in terms of demographic parity, while ensuring that the degradation in the validation AUC does not exceed 3% in comparison to the original biased model. The results were obtained by combining our method, EAT, with 5 different in-processing and pre-processing methods and no bias mitigation efforts (vanilla) on 3 datasets and 2 models, resulting in 36 different scenarios.

| Dataset | Model | Debiasing method | Running time ↓ |
|---|---|---|---|
| Twitter | BERT | Instance weighting (Zhang et al., 2020) | 197% |
| | BERT | CDA (Lu et al., 2020) | 285% |
| | BERT | CDS (Hall Maudslay et al., 2019) | 200% |
| | BERT | Gender blindness (De-Arteaga et al., 2019) | 206% |
| | BERT | EAR (Attanasio et al., 2022) | 208% |
| | BERT | EAT (ours) | **100%** |
| | RoBERTa | Instance weighting (Zhang et al., 2020) | 197% |
| | RoBERTa | CDA (Lu et al., 2020) | 273% |
| | RoBERTa | CDS (Hall Maudslay et al., 2019) | 200% |
| | RoBERTa | Gender blindness (De-Arteaga et al., 2019) | 198% |
| | RoBERTa | EAR (Attanasio et al., 2022) | 200% |
| | RoBERTa | EAT (ours) | **100%** |
| Wikipedia | BERT | Instance weighting (Zhang et al., 2020) | 198% |
| | BERT | CDA (Lu et al., 2020) | 273% |
| | BERT | CDS (Hall Maudslay et al., 2019) | 200% |
| | BERT | Gender blindness (De-Arteaga et al., 2019) | 198% |
| | BERT | EAR (Attanasio et al., 2022) | 203% |
| | BERT | EAT (ours) | **100%** |
| | RoBERTa | Instance weighting (Zhang et al., 2020) | 199% |
| | RoBERTa | CDA (Lu et al., 2020) | 280% |
| | RoBERTa | CDS (Hall Maudslay et al., 2019) | 200% |
| | RoBERTa | Gender blindness (De-Arteaga et al., 2019) | 203% |
| | RoBERTa | EAR (Attanasio et al., 2022) | 203% |
| | RoBERTa | EAT (ours) | **100%** |
| Jigsaw | BERT | Instance weighting (Zhang et al., 2020) | 198% |
| | BERT | CDA (Lu et al., 2020) | 255% |
| | BERT | CDS (Hall Maudslay et al., 2019) | 200% |
| | BERT | Gender blindness (De-Arteaga et al., 2019) | 198% |
| | BERT | EAR (Attanasio et al., 2022) | 202% |
| | BERT | EAT (ours) | **100%** |
| | RoBERTa | Instance weighting (Zhang et al., 2020) | 197% |
| | RoBERTa | CDA (Lu et al., 2020) | 237% |
| | RoBERTa | CDS (Hall Maudslay et al., 2019) | 200% |
| | RoBERTa | Gender blindness (De-Arteaga et al., 2019) | 198% |
| | RoBERTa | EAR (Attanasio et al., 2022) | 200% |
| | RoBERTa | EAT (ours) | **100%** |

Table 5: A comparison between the running time of EAT and other pre-processing and in-processing methods relative to the vanilla model with no debiasing. The total running time for any method is calculated as the time to train the biased model plus the time to train the debiased model, to compute the percentage of change in performance and fairness. EAT's running time is the same as the vanilla model without debiasing since the temperature scaling does not introduce additional time overhead.

## A.2   Packages used

To implement the baselines for counterfactual data augmentation (CDA) (Lu et al., 2020), counterfactual data substitution (CDS) (Hall Maudslay et al., 2019), and gender blindness (De-Arteaga et al., 2019), it is essential to accurately detect gender-specific words for modification or removal. We used the publicly available *gender-bender* Python package[3] for this purpose. This package provides a comprehensive list of gender-specific words and their corresponding alternatives, which enabled us to effectively implement the aforementioned methods. We used the detoxify library[4] for measuring toxicity.

---

[3] https://www.github.com/Garrett-R/gender_bender
[4] https://pypi.org/project/detoxify/

### A.3 Number of trainable parameters

In text classification, our experiments were conducted on BERT (Devlin et al., 2019) and RoBERTa (Liu et al., 2019) base models, which possess 110 and 125 million trainable parameters, respectively. As for text generation, we used GPT-Neo (Black et al., 2021) with 1.3 and 2.7 billion parameters.

### A.4 Infrastructure used

We used a single NVIDIA A100-SXM4-40GB GPU for our experiments.

### A.5 Running time

The computational time for each experiment is proportional to the size of the corresponding dataset. Using a single GPU, the running time for the vanilla model without debiasing was approximately 4 hours for Twitter and BOLD frameworks, whereas it was 12 and 24 hours for the Wikipedia and Jigsaw datasets, respectively.

We also report the computational time for different debiasing methods. EAT reduces bias using a temperature scaling factor in the attention map after the model is trained. This means that our method does not require the model to be re-trained to find a new set of weights. In contrast, pre-processing and in-processing debiasing methods require training the model from scratch with alterations either in the training data (for pre-processing) or the objective function (for in-processing). Table 5 shows how much extra time each debiasing method takes compared to the vanilla model. Specifically, EAT's extra time is negligible because it simply involves adding a temperature scaling factor to the attention map, hence it is be reported as 100% of the vanilla model's time. On the other hand, the rest of the baselines have over 100% values.

### A.6 Dataset imbalance

The percentage of examples with positive labels in the Twitter, Wikipedia, and Jigsaw datasets are 20.29%, 9.62%, and 5.98%, respectively.

### A.7 Decoding configurations for text generation

We applied the following configurations for the text generation using GPT-Neo used in the BOLD experiments:

- The maximum allowed tokens for generation, excluding the prompt tokens is 50 tokens.
- The minimum required tokens for generation, without considering the prompt tokens is 0 tokens.
- We employed sampling, instead of using greedy decoding.
- No beam search was utilized.

## B Results on additional fairness metrics

We present supplementary results that demonstrate the effectiveness of our proposed method, using two additional fairness metrics: equality of odds (EqOdd) and equality of opportunity (Beutel et al., 2017; Hardt et al., 2016). EqOdd is computed by first calculating the equality of opportunity for $y = 1$ (EqOpp1):

$$\begin{aligned}
\text{EqOpp1} = 1 - |p(\hat{y} = 1 | z = 1, y = 1) \\
- p(\hat{y} = 1 | z = 0, y = 1)|,
\end{aligned} \tag{5}$$

and $y = 0$ (EqOpp0):

$$EqOpp0 = 1 - |p(\hat{y} = 1|z = 1, y = 0) \\ -p(\hat{y} = 1|z = 0, y = 0)|, \tag{6}$$

and computing the average:

$$EqOdd = 0.5 \times (EqOpp1 + EqOpp0). \tag{7}$$

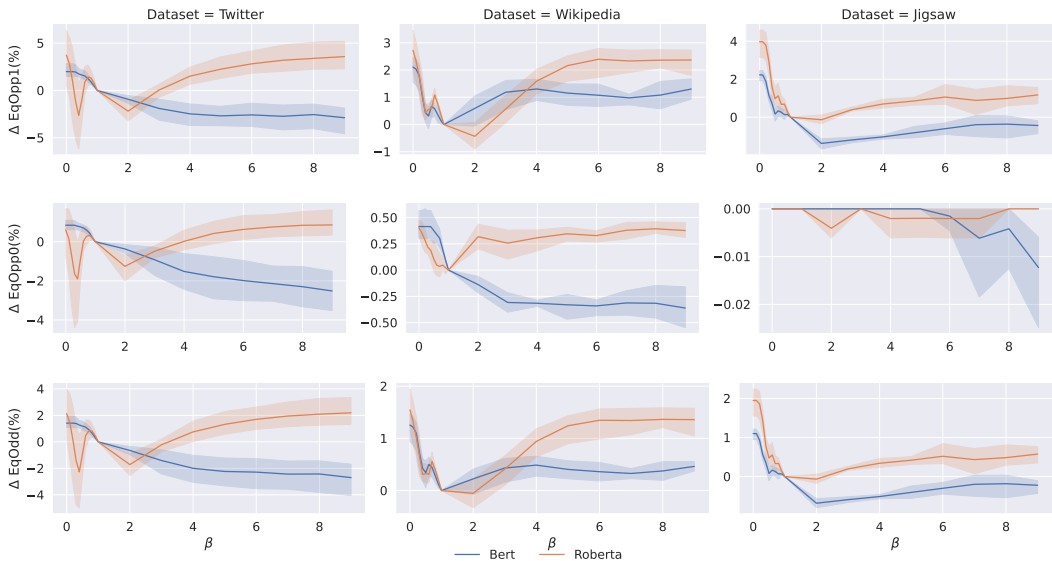

Figure 7: Equality of opportunity (5) (6) and equality of odds (7), using different temperature scaling factors ($\beta$) on the different models and datasets. The values of $\beta$ that are smaller or larger than 1 correspond to maximizing or minimizing the attention entropy, respectively.

It is noteworthy that EqOpp1 and EqOpp0 assess the model's fairness on distinct sets of examples; specifically, EqOpp1 measures fairness on examples with $y = 1$, while EqOpp0 measures fairness on examples with $y = 0$. EqOdd is simply the average of these two metrics. Figure 7 illustrates that both EqOpp1 and EqOdd concur with the DP metric and exhibit similar trends when attention modulation is applied. However, the EqOpp0 fails to improve on the Jigsaw dataset which suggests that the model's bias is primarily concentrated in the $y = 1$ examples (which are labeled as toxic), and thus modulating the attention map improves fairness in this group of examples, but not in the $y = 0$ examples. Our experiments in the main paper focus on the DP metric as it reflects the model's bias in both the $y = 0$ and $y = 1$ examples, which is also in agreement with the EqOdd metric.

## C   Additional results

In this section, we provide additional results on the bias in text classification and generation models.

### C.1   Qualitative results

Table 6 shows the tokens with the highest attention weights in BERT and RoBERTa models on the Wikipedia toxicity detection task using different debiasing methods. EAT puts the highest attention weights on the tokens that are more relevant to the task, as opposed to the vanilla model and EAR, which focus more on gender tokens.

### C.2 Quantitative results

Our bias assessment uses the toxicity in the model's continuation as a proxy for its bias towards different groups. This is in line with the original BOLD paper (Dhamala et al., 2021). Figures 8 and 9 show a detailed comparison for the percentage of change in toxicity when using EAT with $\beta = 0.9$ and random perturbation using GPT-Neo with 1.3 and 2.7 billion parameters, respectively. We used the detoxify library for measuring toxicity. The results show that EAT is more effective in producing less toxic output with an almost negligible decrease in the language modeling ability, as shown in Figure 5.

| Input | Model | Method | Token with highest att. |
|:---:|:---:|:---:|:---:|
| Hello Dan, please fix LanguageRo | | Vanilla | Dan |
| .php first, it has explicit references | BERT | EAR | Dan |
| to Wikipedia. | | EAT | Wikipedia |
| Emilia, do you find more inform- | | Vanilla | Emilia |
| -ation about this Swedish diva? | RoBERTa | EAR | Emilia |
| If you do so, help complete this article! | | EAT | article |

Table 6: Tokens with the highest attention using BERT and RoBERTa on the toxicity detection task using Wikipedia dataset for different bias mitigation methods.

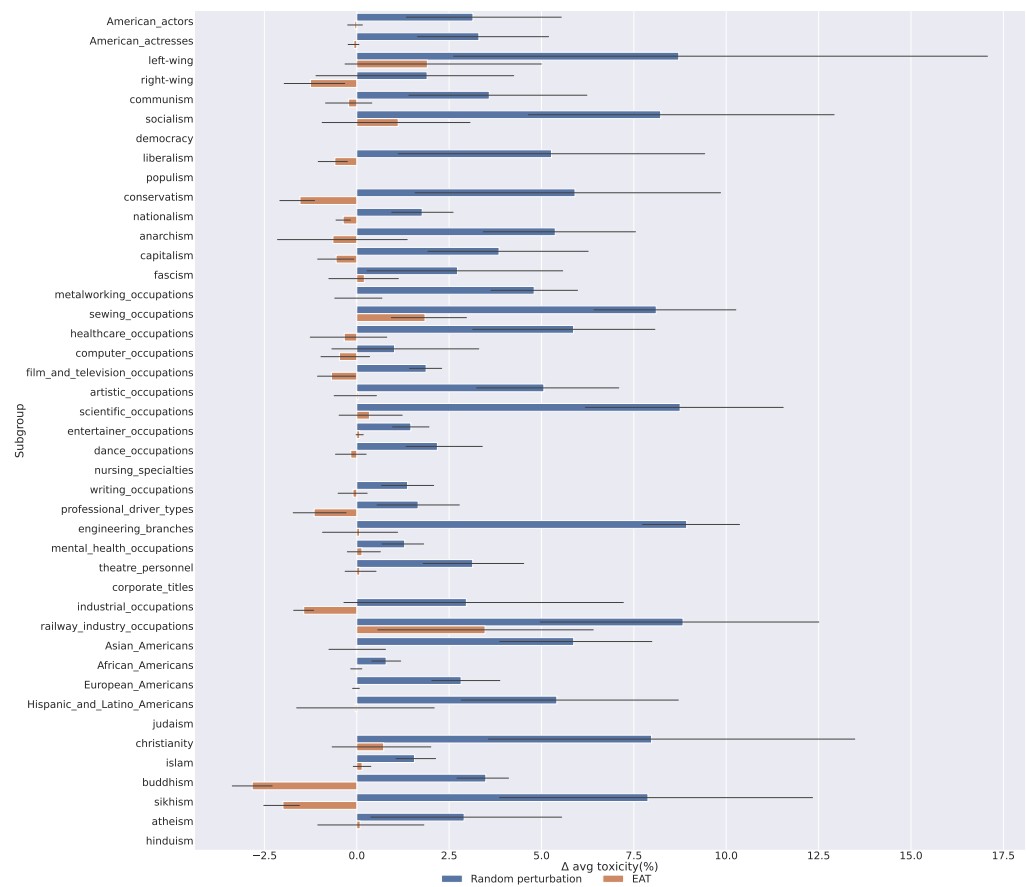

Figure 8: The percentage of change in toxicity when using random perturbation and EAT with $\beta = 0.9$ for GPT-Neo with 1.3 billion parameters, compared to the vanilla model. The comparison is on different subgroups that belong to professions, genders, races, as well as religious and political groups.

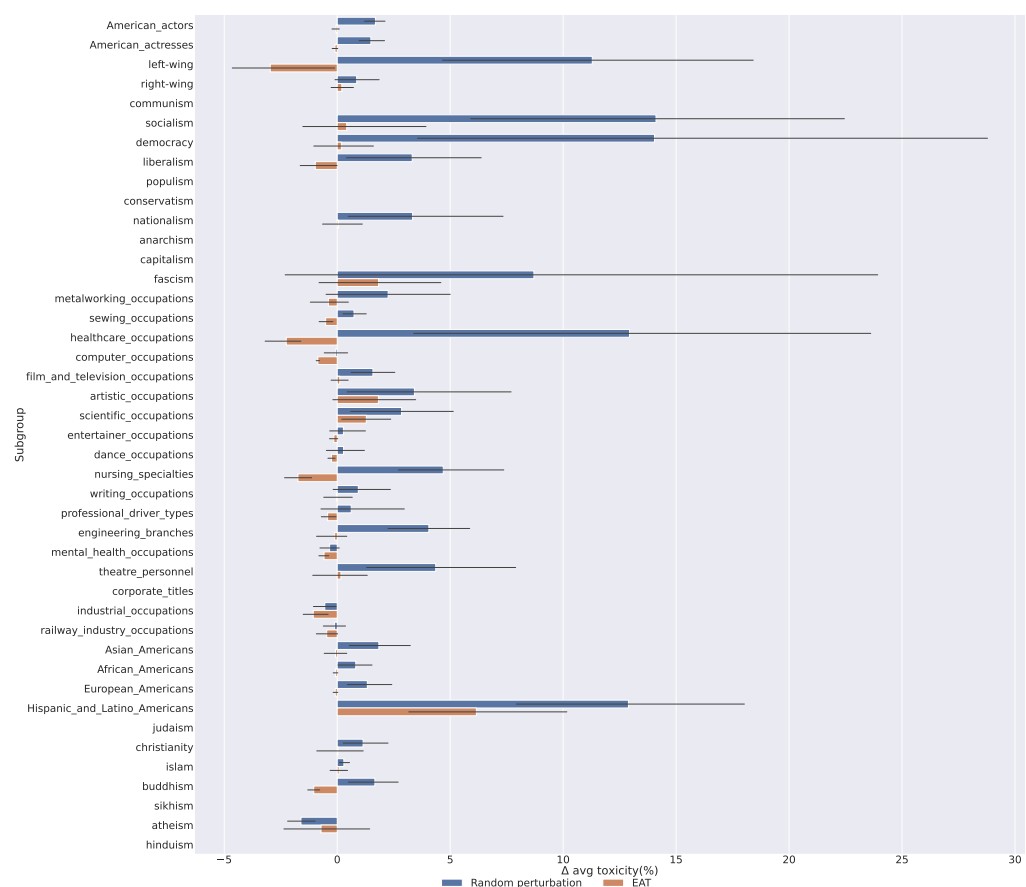

Figure 9: The percentage of change in toxicity when using random perturbation and EAT with $\beta = 0.9$ for GPT-Neo with 2.7 billion parameters, compared to the vanilla model. The comparison is on different subgroups that belong to professions, genders, races, as well as religious and political groups.

