# OpenReview forum: "Should We Attend More or Less? Modulating Attention for Fairness"
_colmweb.org/COLM/2024/Conference — COLM_

### Official Review · Reviewer_vM14 · 2024-05-08

**Rating:** 5
**Confidence:** 3
**Ethics Flag:** 1

**Summary:**

In this work the authors empirically investigate the impact of scaling learned attention weights based on a temperature parameter.
They argue against the previously proposed strategy of maximizing attention entropy in order to improve model fairness, and instead say that minimizing attention entropy may improve fairness, in certain conditions.

I appreciate the thorough experiments and results.
However, I'm having a hard time seeing the significance of this work.
If I understand correctly, the major takeaway is that the answer to "should we minimize or maximize attention entropy?" is "It depends."
This empirical result is certainly useful, especially in contrast to Attanasio et al. where the proposed "fix" is always to maximize entropy.

However, it is unclear when minimization or maximization should be deployed.
What is underneath the fact that for some architectures/datasets/in-processing method, minimization is better?
It would be really interesting to see a detailed analysis of the conditions under which minimization outperforms maximization and vice versa.
Also, examples of the impacts of applying EAT would also be helpful.
Show the attention outputs pre- and post-EAT to demonstrate that the bias is being mitigated.
Otherwise it looks a lot like a fine-tuning exercise that just happens to reduce demographic parity.
Even in Appendix B, there is just a lot of variety in the results, which makes it hard to draw conclusions.

**Reasons To Accept:**

- Thorough evaluations
- Interesting claim that pushes back against prior work

**Reasons To Reject:**

- Results are difficult to parse and not contextualized
- It is unclear why and when minimizing versus maximizing attention entropy is preferred

---

> ### Author Rebuttal · Authors · 2024-05-28
>
> **Thank you for your detailed review and engagement in our work. We are delighted to hear that you consider our work interesting and with thorough evaluations. We address your main points below.**
> >Results are difficult to parse and not contextualized. Even in Appendix B, there is just a lot of variety in the results.
>
> We are sorry about this confusion. The results in Figure 2 and Appendix B demonstrate that fairness is not always maximized by attending to a wider context, as previously claimed by Attanasio et al., (2022), and that the architecture and dataset determine whether the model needs to focus on a wider/narrower context to improve fairness. We will clarify this confusion in the manuscript.
> >It is unclear why and when minimizing versus maximizing attention entropy is preferred
>
> Thanks for bringing up this important point. If the model attends to stereotypical tokens in a narrow context, attention entropy maximization makes the model focus on a wider context, leading to less bias. However, if the model already attends to a wide context containing stereotypical words, attention entropy minimization makes the model focus on a narrower context, which also reduces bias.
>
> >Examples of the impacts of applying EAT would also be helpful. Show the attention outputs pre- and post-EAT to demonstrate that the bias is being mitigated.
>
> This is a great suggestion. We will add more qualitative examples, as shown below.
>
> For toxicity detection using BERT and Input *“Hello Dan, please fix LanguageRo.php first, it has explicit references to Wikipedia.”*:
> | **Method** | **Token with highest attention** |
> |:---:|:---:|
> | Vanilla | Dan |
> | EAT | Wikipedia |
>
> Input *“Freedom will and must prevail, your tyranny will come to an end, Nicholas.”*:
>
> | **Method** | **Token with highest attention** |
> |:---:|:---:|
> | Vanilla  | Nicholas|
> | EAT | tyranny |
>
> Using RoBERTa and Input  *“Emilia, do you find more information about this Swedish diva? If you do so, help complete this article!”*:
> | **Method** | **Token with highest attention** |
> |:---:|:---:|
> | Vanilla | Emilia|
> | EAT | article|
>
> >I'm having a hard time seeing the significance of this work.
>
> As the reviewer had rightly pointed out in their summary, our main contribution is showing empirically and conceptually that the previous work by Attanasio et al., (2022), claiming that maximizing attention entropy always improves fairness, is not general. We will make our main contribution clearer in the revised manuscript.

---

> > ### Comment · Reviewer_vM14 · 2024-06-04
> >
> > Thank you for the response. I appreciate the comments.
> > I'm still at the same point with my scores. The major question for me is: is it enough to say that there are various scenarios where prior work doesn't hold, or are detailed delineations of cases where we minimize/maximize entropy required?

---

> > ### Author Response · Authors · 2024-06-04
> > **Cases where we minimize/maximize entropy**
> >
> > We appreciate your response and engagement.
> >
> > The short answer to your question is: it depends on the data and model used. The longer answer is: For different architectures and datasets, models memorize different shortcuts to solve the task by focusing on an overly narrow/wide context, which leads to bias. The optimal beta mitigates this bias by modulating the model’s attention entropy.

---

### Official Review · Reviewer_UFmf · 2024-05-11

**Rating:** 6
**Confidence:** 3
**Ethics Flag:** 1

**Summary:**

This paper proposes an intra-processing social bias mitigation method that modulates the attention matrix through a validation set. The proposed method is task/model-agnostic and is supposed to be a light-weight solution for the social bias area. However, there are some concerns that should be clarified to improve the paper (please see Weaknesses and Questions below).

**Questions To Authors:**

- It is rather straightforward that the choice of beta can be applied to different layers, implying the knowledge of social bias or spurious correlation is stored among various layers. Do the authors have any insights or empirical observations in this respect?
- Do the authors have any case studies regarding how the choice of beta (or the modulation choice minimization/maximization) is relevant to the linguistic characteristics of a sentence?

**Reasons To Accept:**

- The proposed idea is an intuitive and reasonable approach for intra-processing social bias mitigation.
- The analysis includes additional experimental results for other transformer architectures and tasks.
- The paper is well presented.

**Reasons To Reject:**

- I really appreciate the authors including testing for other transformer architectures and language generation tasks in the Appendix. However, the paper would benefit from including more experimental results using other popular debiasing methods such as Mable [1] and [2]. I do not expect the performance of the proposed method to be better than these two methods, but it is important to include these comparisons so that we can discern any real differences among these different methods.
- The authors show the perplexity results to indicate the influence on the language modeling ability, but additional and more detailed results, for example on GLUE benchmarks or any other downstream classification tasks, are necessary to support the paper’s claims.

[1] He, Jacqueline, et al. "MABEL: Attenuating Gender Bias using Textual Entailment Data." Proceedings of the 2022 Conference on Empirical Methods in Natural Language Processing. 2022.
[2] Ravfogel, Shauli, et al. "Null It Out: Guarding Protected Attributes by Iterative Nullspace Projection." Proceedings of the 58th Annual Meeting of the Association for Computational Linguistics. 2020.

---

> ### Author Rebuttal · Authors · 2024-05-28
>
> **Thank you for your constructive feedback and thoughtful engagement in our work. We are motivated to hear that you consider our work well-presented, intuitive, and with thorough analyses. We address your main points below.**
> >the paper would benefit from including more experimental results using other popular debiasing methods such as Mable [1] and [2].
>
> This is a very interesting suggestion. We added MABLE as a baseline and launched the experiments comparing it to EAT. If we get the results before the rebuttal deadline, we will discuss them, but we will add them to the manuscript in any case. Given that MABLE and Nullspace Projection are in-processing and pre-processing methods, respectively, they are both complementary to our proposed post-processing method (and hence can be combined with it).
> > GLUE benchmarks or any other downstream classification tasks, are necessary.
>
> We launched experiments to measure the effect of attention modulation on downstream tasks. We will discuss the results if we get them before the rebuttal period ends, and we will add them to the manuscript in any case.
> > the choice of beta can be applied to different layers, implying the knowledge of social bias or spurious correlation is stored among various layers.
>
> This is an excellent comment. Previous work has shown that the first layers are more critical as they learn more abstract concepts [1], but intuitively any layer could learn bias. This motivates varying the attention modulation based on the layer, which comes at the cost of rendering the hyperparameter search more difficult. We kept the attention modulation factor the same in all layers for simplicity.
>
> [1] Sajjad et al. On the effect of dropping layers of pre-trained transformer models. Computer Speech & Language, 2023.
> >Do the authors have any case studies regarding how the choice of beta is relevant to the linguistic characteristics of a sentence?
>
> This is an interesting question. Based on our understanding, the choice of beta does not depend on the linguistic characteristic of the sentence per se, it rather depends on the model’s attention to stereotypical tokens. If the model attends to stereotypical tokens in a narrow context, the optimal beta would be smaller than 1, making the model focus on a wider context. However, if the model already attends to a wide context containing stereotypical words, the optimal value of beta would be larger than 1, to make the model focus on a narrower context.

---

> > ### Author Response · Authors · 2024-06-04
> > **The effect of attention entropy modulation on downstream tasks in text generation models**
> >
> > Thanks for your patience. We have obtained the results showing the effect of attention entropy modulation on 7 different downstream tasks using GPT-Neo. The values for all tasks refer to accuracy scores, except for the DROP task which uses F1 score.
> > | **Beta\Task** | **ARC (0-shot)** ↑| HellaSwag (0-shot) ↑| MMLU (0-shot) ↑| TruthfulQA (0-shot) ↑| Winogrande (0-shot) ↑| GSM8K (5-shot) ↑|DROP (0-shot) ↑|
> > |:---:|:---:|---|---|---|---|---|---|
> > | 0.7 (attention entropy max.) | 0.2201 ± 0.0121 | 0.2913 ± 0.0045 | 0.2290 ± 0.0035 | 0.3430 ± 0.0108 | 0.5280 ± 0.0140 | 0.0129 ± 0.0031 | 0.0206 ± 0.0008 |
> > | 0.8 (attention entropy max.) | 0.2261 ± 0.0122 | 0.2960 ± 0.0046 |  0.2293 ±  0.0035 | 0.3496 ± 0.0108 | 0.5114 ± 0.0140 | 0.0182 ±  0.0037 | 0.0197 ± 0.0007 |
> > | 0.9 (attention entropy max.) | 0.2287 ± 0.0123 | 0.3009 ± 0.0046 | 0.2302 ± 0.0035 | 0.3511 ± 0.0108 | 0.5099 ± 0.0140 | 0.0136 ± 0.0032 | 0.0199 ± 0.0007 |
> > | **1 (no attention entropy modulation)** | **0.2312 ± 0.0123** | **0.3040 ± 0.0046** | **0.2323 ± 0.0036** | **0.3570 ± 0.0109** | **0.5043 ± 0.0141** | **0.0159 ± 0.0034** | **0.0211 ± 0.0008** |
> > | 1.1 (attention entropy min.)| 0.2321 ± 0.0123 | 0.3051 ± 0.0046 |  0.2318 ± 0.0036 | 0.3580 ± 0.0109 | 0.5059 ± 0.0141 | 0.0136 ± 0.0032 | 0.0197 ± 0.0008 |
> > | 1.2 (attention entropy min.)| 0.2329 ± 0.0124 | 0.3053 ± 0.0046 | 0.2305 ± 0.0035 | 0.3609 ± 0.0109 | 0.5122 ± 0.0140 |0.0220 ± 0.0040 | 0.0191 ± 0.0008 |
> > | 1.3 (attention entropy min.)|0.2304 ± 0.0123 | 0.3049 ± 0.0046 | 0.2295 ± 0.0035 | 0.3611 ± 0.0109 | 0.5036 ± 0.0141 | 0.0174 ± 0.0036 | 0.0185 ± 0.0007 |
> >
> > The results show that attention entropy modulation (*i.e.* maximization or minimization) does not lead to substantial degradation on the downstream tasks used, while improving the performance in certain cases.
> > More specifically, attention entropy minimization leads to better performance on ARC, HellaSwag, TruthfulQA, GSM8K, and Winogrande; while attention entropy maximization improves performance on GSM8K and Winogrande. We will add these results to the manuscript, and we thank the reviewer for the constructive suggestion.

---

> > > ### Author Response · Authors · 2024-06-04
> > > **Results of MABLE baseline suggested by the reviewer**
> > >
> > > We also obtained the results for the in-processing baseline suggested by the reviewer: MABLE. Our results show that MABLE consistently improves fairness, and its effect is amplified when combined with our proposed post-processing method: EAT. The results are averaged over 5 seeds and use demographic parity as the gender fairness metric.
> > >
> > >  | **Dataset** | **Model** | **Method** | **DP ↑** |
> > >  |:---:|:---:|:---:|:---:|
> > >  | Twitter | BERT | Vanilla | 0.955 |
> > >  | Twitter | BERT | MABLE| 0.965 |
> > >  | Twitter | BERT | EAT (ours) | 0.993  |
> > >  | **Twitter** | **BERT** | **MABLE+EAT**| **0.994** |
> > >  | | | | | |
> > >  | Twitter | RoBERTa | Vanilla | 0.958 |
> > >  | Twitter | RoBERTa | MABLE| 0.984 |
> > >  | Twitter | RoBERTa | EAT (ours) | 0.967 |
> > >  | **Twitter** | **RoBERTa** | **MABLE+EAT**| **0.992** |
> > >  | | | | |
> > > | Wikipedia | BERT | Vanilla | 0.975 |
> > > | Wikipedia | BERT | MABLE| 1.00 |
> > > | Wikipedia | BERT | EAT (ours)| 1.00 |
> > > | **Wikipedia** | **BERT** | **MABLE+EAT**| **1.00** |
> > > | | | | |
> > > | | | | |
> > > | Wikipedia | RoBERTa | Vanilla | 0.974 |
> > > | Wikipedia | RoBERTa | MABLE| 0.998 |
> > > | Wikipedia | RoBERTa | EAT (ours) | 0.997 |
> > > | Wikipedia | RoBERTa | **MABLE+EAT**| **0.999** |
> > > | | | | |

---

### Official Review · Reviewer_czv4 · 2024-05-13

**Rating:** 7
**Confidence:** 4
**Ethics Flag:** 1

**Summary:**

This paper proposes a novel intra-processing method, Entropy-based Attention Temperature Scaling (EAT), to modulate attention entropy for improving fairness in NLP models. The problem is that transformer models memorize stereotypical tokens. EAT does temperature scaling after training. It adds noise to the softmax in the attention mechanism.

The study uses three datasets: Twitter, Wikipedia, and Jigsaw, all in English, for two text classification tasks,  sexism and toxicity detection. Performance goes down in the overall performance, but it reduces disparities in AUC between messages that refer to various groups.

**Reasons To Accept:**

* Interesting idea for a method.
* The computational benefit is not requiring pre-training or fine-tuning of a model.
* The paper demonstrates usefulness in the three tasks.

**Reasons To Reject:**

* The method is arguably not a great advance over previous work with temp scaing.
* It is not clear how to trade off the decreased performance and improved AUC equality score.
* Optimal parameters vary a lot across datasets, so it would not be useful with data that is not labeled for the sensitive attributes.
* The datasets and metrics used seem specific -- the choices could have been motivated more and compared to alternatives.

---

> ### Author Rebuttal · Authors · 2024-05-28
>
> **Thank you for your detailed review and comments. We are encouraged that you find our work interesting, empirically useful, and computationally efficient. We address your main points below.**
> >The method is arguably not a great advance over previous work with temp scaing.
>
> We agree that the idea of temperature scaling exists already in the literature, as Reviewer q9VN  and czv4 had rightly pointed out. Our main contribution lies in demonstrating, both empirically and conceptually, that attention entropy maximization does not always lead to fairer models, as previously claimed by Attanasio et al., (2022).
> >It is not clear how to trade off the decreased performance and improved AUC equality score.
>
> Thanks for bringing this up. As mentioned in Section 5.4, we maximize fairness while ensuring that the performance drop does not exceed 3% on the validation dataset.
> >Optimal parameters vary a lot across datasets, so it would not be useful with data that is not labeled for the sensitive attributes.
>
> We agree that the optimal value of the temperature scaling hyperparameter varies based on the setting. For different architectures and datasets, models memorize different shortcuts to solve the task by focusing on an overly narrow/wide context, which leads to bias. The optimal beta mitigates this bias by modulating the model’s attention.
> >The datasets and metrics used seem specific -- the choices could have been motivated more and compared to alternatives.
>
> Thanks a lot for your comment. We chose the datasets and metrics that are widely used in the fairness literature [1-4]. We will add this information to the manuscript to make it clear.
>
> [1] Zhang et al. Demographics Should Not Be the Reason of Toxicity: Mitigating Discrimination in Text Classifications with Instance Weighting. ACL, 2020.
>
> [2] Reddy et al. Benchmarking bias mitigation algorithms in representation learning through fairness metrics. NeurIPS Datasets and Benchmarks Track, 2021.
>
> [3] Shen et al. Optimising Equal Opportunity Fairness in Model Training. NAACL, 2022
>
> [4] Dixon et al. Measuring and mitigating unintended bias in text classification. AIES, 2018.

---

> > ### Comment · Reviewer_czv4 · 2024-06-05
> >
> > Thank you for the clear response.

---

### Official Review · Reviewer_q9VN · 2024-05-13

**Rating:** 6
**Confidence:** 4
**Ethics Flag:** 1

**Summary:**

This paper focuses on minimizing social bias in language models. Specifically, it builds on prior work by Attansio et al., (2022) and attempts to provide an alternative solution to building an accurate language model that also improves fairness metrics. Attansio et al. posit that maximizing the entropy of the attention weights distribution during fine-tuning leads to less biased models (called EAR). This paper argues that this is not necessarily always true; instead it proposes to modulate the attention entropy using a hyperparameter (temperature scaling factor), which could lead to either an increase of decrease in entropy, arguing that the modulation is a factor of the dataset and architecture (henceforth, EAT).

This hypothesis is experimentally validated on three classification datasets (Wikipedia, toxic v/s non-toxic classification; Jigsaw, toxic v/s non-toxic classification; Twitter, sexist v/s non-sexist comment classification) and one open ended generation dataset (BOLD). A variety of alternative and previously proposed approaches are considered as baselines. The quality of the models are measured by AUC for classification and perplexity for open-ended generation, while fairness is measured using Demographic Parity for gender bias and pinned AUC equality difference for other biases (e.g. religion). Findings reveal that the choice of the temperature scaling factor can indeed affect fairness metrics on classification tasks, with EAT consistently improving these metrics compared to EAR. EAT more frequently leads to improved fairness metrics compared to other baselines as well.  On text generation, EAT outperforms the random perturbation baseline on both quality and fairness metrics.

Overall, I like that this work proposes a post-training (or what is called as *intra-processing* in this paper) approach to mitigating fairness issues, compared to Attansio et al., which requires fine-tuning the underlying model. This makes the approach fairly easy to try. I also appreciate that this work does not limit itself to classification tasks. With the direction the community is heading in, it is paramount to measure and mitigate biases in language modeling tasks as well. Finally, I also appreciate that this work attempts to provide a generalization to a prior, well-recognized solution to the problem (i.e. entropy maximization is not sufficient, and modulating the entropy is a more general solution)

However, I do think this work is not technically exciting as it proposes a straightforward application of temperature scaling, which has been used in knowledge distillation, calibration of neural networks [1], etc. Another drawback of this approach is that it requires a per dataset approach to mitigating bias (by identifying the right temperature scaling parameter). Finally, I also think that this paper is missing some side-by-side comparison of examples where EAT improves upon EAR. In the introduction to section 4, the authors argue that *"If there are stereotypical tokens in the narrower context, then attending to a broader context is likely to improve fairness. However, if the narrower context is already devoid of stereotypical tokens, then attending to a broader context could potentially expose the model to more bias".* I think having some examples that actually validate this will strengthen the paper.


[1] - https://arxiv.org/pdf/1706.04599

**Questions To Authors:**

* Citations are incorrectly parenthesized throughout the paper, and this needs fixing [e.g. intra-processing methods Savani et al. (2020) --> intra-processing methods (Savani et al., 2020)].
* Why did you choose the bias metrics that you did in section 5.3? How confident are you that a single bias/fairness metrics supports the findings in this paper? If other fairness metrics are considered, do you expect the findings in this work to generalize?

**Reasons To Accept:**

* This paper proposes an approach to mitigating bias that does not require fine-tuning of the underlying model
* Experiments are not restricted to classification; open-ended text generation results indicate that this method is broadly applicable to other tasks as well.
* Provides an argument that improves upon prior work by making it more general.

**Reasons To Reject:**

* The temperature scaling hyperparameter indicates that this approach needs to be configured for different data distributions/architectures, which makes it less broadly useful, when test data distribution is unknown.
* The technical novelty in this paper is limited as the approach is an application of temperature scaling
* Qualitative insights to back up quantitative results are completely missing.

---

> ### Author Rebuttal · Authors · 2024-05-28
>
> **Thank you for your time in providing a thorough and complete review. We appreciate your interest in understanding all the details in our work and address your main points below.**
>
>
> >The temperature scaling hyperparameter indicates that this approach needs to be configured for different data distributions/architectures.
>
> We agree that one of the limitations of our method is that it assumes having access to validation data from the same distribution as the test data to choose the optimal temperature scaling.
>
> >The technical novelty in this paper is limited as the approach is an application of temperature scaling
>
> We agree that the idea of temperature scaling exists in the literature, as the reviewer rightly pointed out. The main contribution of our work is challenging the existing claim that maximizing attention distribution entropy always improves fairness, by showing empirically and conceptually that this is not a general observation.
>
> >Qualitative insights are completely missing.
>
> Thanks for pointing this out. Table 3 in Section 5.4 provides some qualitative results, but we will add more examples to strengthen the paper, as shown below.
>
>
> For toxicity detection using BERT and Input *“Hello Dan, please fix LanguageRo.php first, it has explicit references to Wikipedia.”*:
>
> | **Method** | **Token with highest attention** |
> |:---:|:---:|
> | Vanilla | Dan |
> | EAR | Dan |
> | EAT | Wikipedia |
>
> Using RoBERTa and Input *“Emilia, do you find more information about this Swedish diva? If you do so, help complete this article!”*:
>
> | **Method** | **Token with highest attention** |
> |:---:|:---:|
> | Vanilla | Emilia|
> | EAR | Emilia|
> | EAT | article|
>
> >How confident are you that a single bias/fairness metrics supports the findings in this paper
>
> In line with related works [1-3], we use four bias assessment metrics for text classification. For gender bias, we use demographic parity, equality of odds, and equality of opportunity; and for general social biases, we use pinned AUC. Due to space limitations, some results are shown in Appendix B. We also use BOLD benchmark for bias assessment in text generation.
>
> [1] Reddy et al. Benchmarking bias mitigation algorithms in representation learning through fairness metrics. NeurIPS Datasets and Benchmarks Track, 2021.
>
> [2] Shen et al. Optimising Equal Opportunity Fairness in Model Training. NAACL, 2022
>
> [3] Dixon et al. Measuring and mitigating unintended bias in text classification. AIES, 2018.

---

> > ### Comment · Reviewer_q9VN · 2024-06-04
> >
> > Thank you for your response. I think adding the qualitative examples will help the paper. Overall, I'll still stand by my initial review that the paper has limited technical novelty, but I appreciate the refutation of prior work by finding a more general approach that solves the underlying problem.

---

> > > ### Author Response · Authors · 2024-06-04
> > > **Thank you**
> > >
> > > Thank you for your time and effort in reviewing our work.

---

### Decision · Program_Chairs · 2024-07-10

**Decision:**

Accept

**Comment:**

This paper reevaluates a previously recognized solution to bias-mitigation, resulting in a very different conclusion (relating to attention entropy, namely that increasing or decreasing entropy can improve fairness, depending on task and dataset). This is an interesting, and the approach taken simple and easily applied without fine-tuning; moreover the experiments were not limited to classification only, which is common in fairness papers. However the paper is lacking any deep insights to explain the key finding - that is, what is the underlying cause of the need to sharpen or smooth the attention, in terms of the data/task/linguistic context/etc. The author response is not satisfying, just summarizing the observed model behavior (see response to vM14). In other respects, the paper is strong: well written, evaluation sound, and the authors added new evaluations in their response.